

# External morphology of eyes and Nebenaugen of caridean decapods–ecological and systematic considerations

Magnus L. Johnson[1], Nicola Dobson[1] and Sammy De Grave[2]

[1] Centre for Environmental and Marine Sciences, University of Hull, Scarborough, UK
[2] Oxford University of Natural History, Oxford, UK

## ABSTRACT

Most caridean decapods have compound eyes of the reflecting superposition kind, and additionally some possess an accessory eye-like organ of unknown function, also referred to as the nebenauge. We examined 308 caridean genera to assess the general morphology of the eye, rostrum length, eye diameter and the presence or absence and, when present, the diameter of the nebenauge. We have attempted to relate these data to ecological and taxonomic considerations. We consider there to be 6 distinct eye types based on the margin between the eyestalk and cornea. The presence of nebenaugen appears to be generally linked to an active lifestyle, as evidenced by the fact that species that have nebenaugen tend to have larger eyes and are more likely to have a distinct rostrum. We suggest that the inconsistencies in its presence/absence under both systematic and ecological lenses may indicate that when present it has various roles relating to behavioural and physiological rhythms.

## INTRODUCTION

### Decapod eyes

The laws of physics impose strict regulation on the optical and physical design of eyes and their limitations are thus manifest in their physical dimensions. Larger eyes generally have greater sensitivity both spatially and with regard to light intensity (*Laughlin, 2001*). In species such as long-bodied decapods that generally have super-position compound eyes, physiological evolution would in theory encourage the development of large, perfectly formed hemispheres on long eyestalks that would give the bearer highly sensitive all round vision and sufficient resolution to enable accurate triangulation. However, in the real world the evolution of eyes and other sensory organs is constrained by factors not directly related to their function such as energy requirements, physical size, information processing limitations (*Laughlin, de Ruyter van Stevenwick & Anderson, 1998*; *Weibel, 2000*), their use as honest signals (*David et al., 2000*) and camouflage requirements (*Gaten, Shelton & Herring, 1992*; *Shelton, Gaten & Herring, 1992*; *Johnson et al., 2000*).

Corresponding author
Magnus L. Johnson,
m.johnson@hull.ac.uk

The eyes of decapods have received considerable attention with regard to their design (*Cronin, 1986*; *Meyer-Rochow, 2001*; *Land & Nilsson, 2002*) and their systematic importance (*Fincham, 1980*; *Gaten, 1998*; *Porter & Cronin, 2009*). The majority of shrimp-like decapods are considered to have superposition optics, thought to be an early superior development of apposition optics that, when dark adapted, enhance the sensitivity of the eye considerably by super-imposing light from a single incident source that reaches several facets onto a single rhabdom (*Land & Fernald, 1992*). This is quite different from an apposition eye, where light from a point source will only stimulate one receptor via a single facet. The superposition design must be of considerable advantage in low light conditions or at depths beyond the reach of down-welling light where bioluminescence is common (*Herring et al., 1990*; *Haddock, Moline & Case, 2010*).

Consideration of eye design in many animals has shed light on the habits of difficult to observe species such as those found in the deep-sea realm or of a crepuscular or nocturnal habit (*Marshall, 1979*; *Herring et al., 1990*). *Welsh & Chace (1937)* and *Welsh & Chace (1938)* examined the eyes of pelagic crustaceans and noted little more than that their large size indicated that they must retain some important function. *Hiller-Adams & Case (1984)* and *Hiller-Adams & Case (1985)* carried out some more detailed interspecific comparisons of some euphausiids and benthic decapods eyes. They noted that benthic decapods eyes get larger with depth while those of (pelagic) euphausiids become smaller. A similar observation was made of crabs by *Eguchi, Dezawa & Meyer-Rochow (1997)*, who found that the eyes of the deep sea crab *Paralomis multispina* were almost twice as large as the similar shallow water species *Petrolisthes* sp.

More recent investigations into the structure of eyes of species from the meso-pelagic realm revealed the tuning of eyes specific to the depths at which they were found and that were well related to the spectral, temporal and spatial availability of light (*Shelton, Gaten & Herring, 1992*; *Frank & Widder, 1999*; *Johnson et al., 2000*; *Johnson, Gaten & Shelton, 2002*; *Mylinski, Frank & Widder, 2005*). For example, the angular distribution of reflective tapetal pigments that enhance sensitivity has been found to be related to the depths that each species inhabits. Those that live beyond the depth at which down-welling light is of no relevance to vision have a complete tapetum, while those found at shallower depths have tapetal distributions that reflect the angular distribution of downwelling light that they experience at the top of their depth range (*Shelton, Gaten & Herring, 1992*; *Johnson et al., 2000*). Absences of tapetal pigment in lateral parts of the eye are thought to be linked to the fact that shrimps 'blink' during the characteristic tail flip escape response (*Shelton et al., 1999*). More recently, close relationships have been found between eye design and habit of commensal shrimps, such that it is possible to determine the lifestyle of an animal from superficial features of the eye (*Dobson, De Grave & Johnson, 2014*).

Examination of the relationship between eye size and body size in vertebrates has suggested that there is a generally logarithmic relationship between body weight and eye diameter but that the precise relationship varies depending on lifestyle (*Howland, Merola & Basarab, 2004*). Leuckart's law suggests that larger, faster moving animals should have larger eyes (*Laughlin, 2001*). Predatory birds, for example, have larger eyes

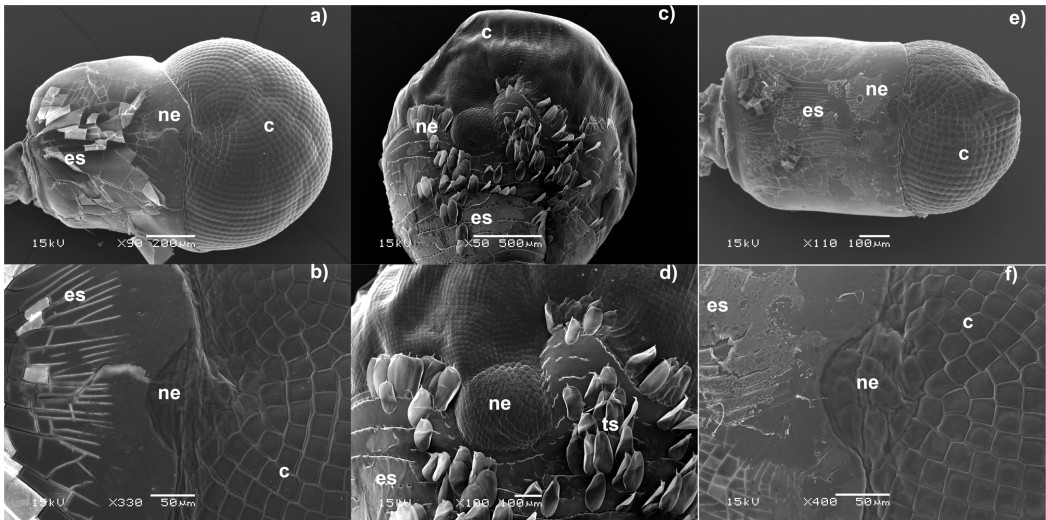

**Figure 1** Electron micrographs of the dorsal region of Caridean eyes. (A–B) *Cuapetes elegans*, (C–D) *Cinetorhynchus hendersoni*. Note the presence of tegumental scales on this species (*De Grave & Wood, 2011*) (E–F) *Gnathophyllum americanum*. Note the peaked cornea which is a typicalfeature of this genus. ed, eye stalk; ne, nebenauge; c, cornea.

than passerines (*Brooke, Hanley & Laughlin, 1999*) and reef fish split into four general types (nocturnal/diurnal v grazer/predator) according to the relative sizes of their eyes and mouths (*Goatley & Bellwood, 2009*). In scarabid beetles, data extracted from *Gokan & Meyer-Rochow (2000)* suggest that absolute eye size (ommatidial length) is larger in nocturnal that diurnal species, and *Meyer-Rochow & Gál (2004)* found that there are structural constraints that severely limit performance in species with compound eyes of less than 0.5 mm diameter.

## The nebeneuge

There are a bewildering array of additional light receptors in decapods—various arrangements of frontal eyes, the nebenaugen and even abdominal eyes (*Elofsson, 2006*). The small size of these extra 'eyes' suggests that they have restricted/limited functions and act as neural filters for specific bits of information.

The nebenauge of decapods (Fig. 1) has also been referred to as doppelauge, a deeply pigmented ocellus, dorsal pigment spot, the pigment black spot or ocellus and accessory compound eye (*Itaya, 1976*; *Gaten, Shelton & Herring, 1992*; *Ugolini & Borgioli, 1993*). The precise arrangement and morphology of the nebeuge varies between species but it superficially resembles an apposition compound eye, closely associated with the eye proper, embedded in the eye stalk with clearly hexagonal facets and sometimes having a slightly convex cross-section. It consists of a discrete region of cornea, often positioned dorsally or posteriorly at the margin of the eye. The facets are hexagonal or irregular, even in cases where the rest of the eye has square facets. The nebenauge can vary in size from small (only a few facets in diameter) to something that can occupy about 1/5 of the diameter of the eyestalk. It is generally mentioned in taxonomic papers if it is present, but appears to be assigned little in the way of systematic importance, with the exception of

*Komai (1999)* who uses its presence/absence as a diagnostic characteristic to differentiate between *Pandalus* and *Atlantopandalus*.

The internal structure of the nebenauge resembles that of the eye proper, with obvious ommatidia and possessing rhabdoms a doptric layer, cornea and facets. However *Itaya (1976)* suggested that the angular divergence of the ommatidia of the crustacean nebenauge indicated that it was not image forming. *Carlisle (1959)* tentatively suggests that the nebenauge is connected to the sensory papilla x-organ in *Pandalus borealis* and a diagram in *Carlisle & Knowles (1959)* clearly indicates a link between the nebenauge and the sinus gland in *Palaemon serratus* but no suggestion is made as to its possible function. However, they do point out that light appears to have some impact on the release of hormones from the sinus gland in this species. *Brown (1961)* investigating the decapod *Hippolyte varians* which has no nebenaugen noted that it still exhibits a daily fluctuation in colour, even when kept in constant darkness, and integumental pigmentation is regulated by eyestalk hormones. Removal of eyestalks abolishes the daily rhythm and it is very clear that light, amongst other environmental factors, has an impact on timing of reproduction and the moult cycle via a complex network of neuropeptides and the sinus gland complex (*Nagaraju, 2011*). Although hormones and factors deriving from the eyestalk are known to control growth, moulting, heart rate, metabolic rate, metabolism of sugars and proteins, water balance, pigment movements and reproduction (*Adiyodi & Adiyodi, 1970*), to date the details of the mechanisms of reproductive regulation and moult cycle progression remain hazy. Based on examinations of the eyes of mesopelagic species, *Gaten, Shelton & Herring (1992)* suggest that the nebenauge is directed at the region of highest light intensity and may have a stabilising role or have something to do with the regulation of diurnal vertical migrations observed in such species.

The function and systematic relevance of the nebenauge of carideans remains unclear (*Gaten, Shelton & Herring, 1992*; *Ugolini & Borgioli, 1993*) despite a long history of interest in the structure of the main eye and the physiological importance of the closely associated sinus gland complex. Also, there have been no extensive examinations of the variation in gross eye design amongst caridean decapods, and few decapod systematicists recognise its potential as a useful diagnostic character.

### Aims

The aims of this investigation are to examine the potential taxonomic significance and likely ecological role of the nebenauge of caridean shrimps through an extensive examination of its presence, absence and relative size in representatives of every genera of caridean. We also assess the variation in gross eye structure of decapods generally and in relation to the presence/absence of the nebenauge.

## METHODS

A total of 179 adult decapod genera were accessed from the Oxford University Natural History Museum collection. Animals had generally been preserved by immersion in 75% ethanol. One representative of every genus available in the collection was used. Specimens were mounted dorsal-side up under a dissecting microscope, and their eyes sketched using
a drawing attachment. Measurements of carapace length, rostrum length, eye diameter, and when present nebenauge diameter were taken using a calibrated eyepiece scale. For large specimens, carapace length was measured using callipers. Carapace length was taken as the measurement from the posterior edge of the eye socket to the dorsal-posterior edge of the carapace. Eye diameter was measured as the maximum anterio-posterior length across the eye and nebenauge diameter was taken as the measurement at the widest point. Rostrum length was taken as the straight line from the tip of the rostrum to the posterior-most edge of the eye socket.

Although troglobitic genera were examined, we found they do not have well-developed eyes and so they were excluded from our analysis. Where there were several genera to choose from in the collection, we examined the largest representative or used a species where there were numerous individuals available.

Details of a further 133 genera were extracted from the literature. For the most part, the relevant taxonomic papers were found after consultation with *De Grave & Fransen (2011)*. When possible, carapace length, rostrum length, nebenauge presence were noted, either by direct reference to the text or through observations taken from the drawings. Preliminary analysis of the dataset (see Supplemental Information) indicated that measurements taken from papers were not robust enough to be anything more than indicative and so were only used for presence/absence of nebenaugen and rostrum length classification. Analysis of a dataset with date since collection of samples indicated no effect of sample age on superficial eye morphology (see Supplemental Information).

Apportioning precise depth ranges on the basis of a few samples would not be wise, so animals were classified according to their deepest recorded depth distribution into freshwater, coastal (0–99 m), shelf (100–300 m) and deep (>300 m) based on available information from the literature. Freshwater species were not classified by depth because of the wider range of optical properties of lentic and lotic environments. Animals were also categorised by habitat (temperate, tropical, deepsea) and as either commensal or free living.

## Analysis

All analyses were carried out using the statistical package R (*Ihaka & Gentleman, 1996*). Where necessary and possible, data were normalised via log transformations so that parametric techniques could be applied. Normality was assessed using the Shapiro–Wilks test.

## RESULTS

### Morphological variations

We found that eyes could generally be classed into 6 major types according to the shape of the margin between the eyestalk and the cornea (Fig. 2), general shape and further classified by the presence or absence of the nebenauge.

The distribution of eye types among families was not random ($X^2 = 285.5$, d.f. $= 11,308$, $p < 0.0001$) and neither was the distribution of nebenaugen among them (See Table 1). Although present in 14 families, the most basic eye design (Type 1) which consists of a tube shaped eyestalk and a simple hemispherical eye was the most common

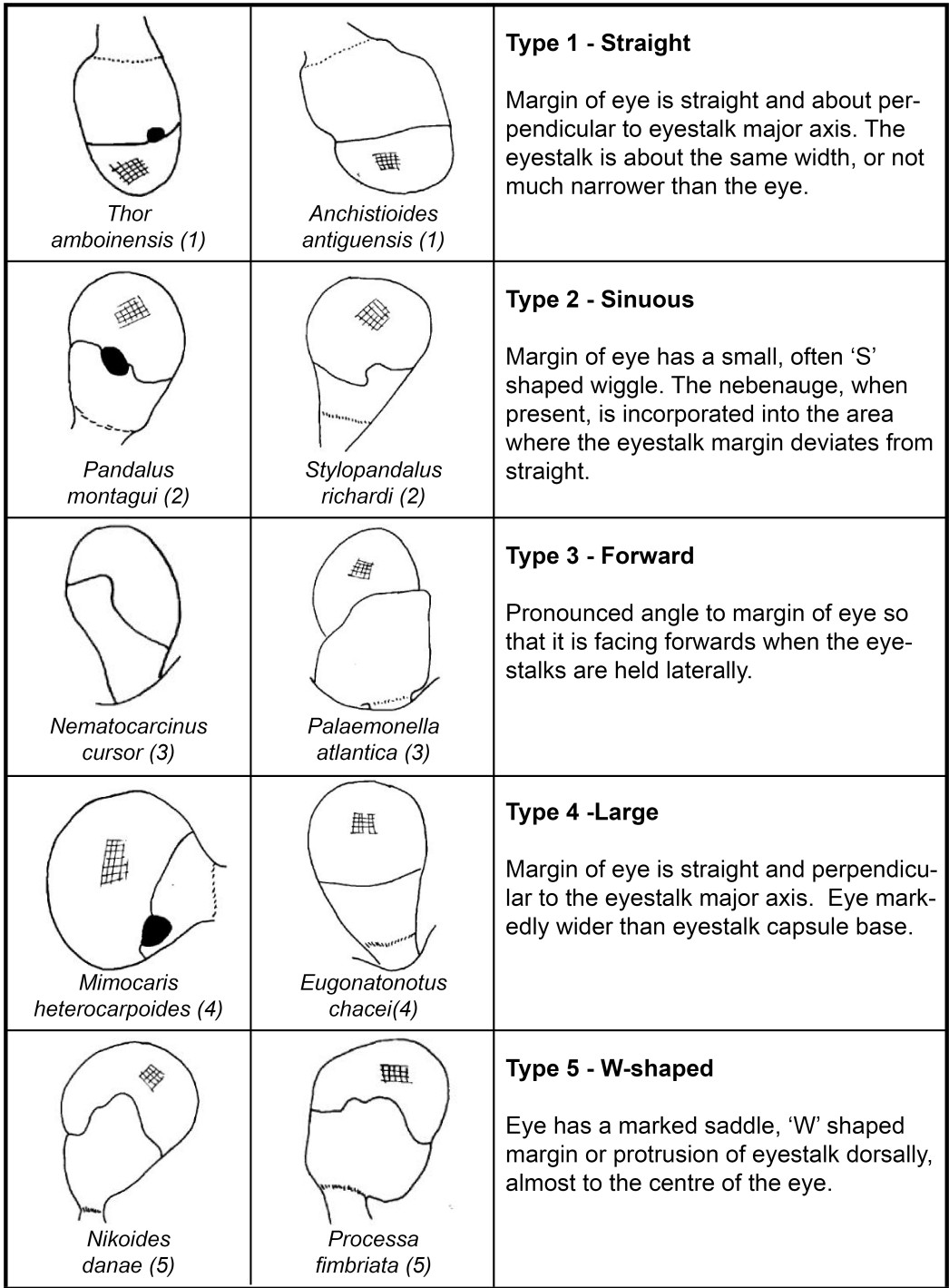

**Type 1 - Straight**

Margin of eye is straight and about perpendicular to eyestalk major axis. The eyestalk is about the same width, or not much narrower than the eye.

*Thor amboinensis (1)*

*Anchistioides antiguensis (1)*

**Type 2 - Sinuous**

Margin of eye has a small, often 'S' shaped wiggle. The nebenauge, when present, is incorporated into the area where the eyestalk margin deviates from straight.

*Pandalus montagui (2)*

*Stylopandalus richardi (2)*

**Type 3 - Forward**

Pronounced angle to margin of eye so that it is facing forwards when the eyestalks are held laterally.

*Nematocarcinus cursor (3)*

*Palaemonella atlantica (3)*

**Type 4 -Large**

Margin of eye is straight and perpendicular to the eyestalk major axis. Eye markedly wider than eyestalk capsule base.

*Mimocaris heterocarpoides (4)*

*Eugonatonotus chacei(4)*

**Type 5 - W-shaped**

Eye has a marked saddle, 'W' shaped margin or protrusion of eyestalk dorsally, almost to the centre of the eye.

*Nikoides danae (5)*

*Processa fimbriata (5)*

**Figure 2** **Morphological types of eyes based on the corneal margin when viewed dorsally.** Type 6 (not illustrated) includes all reduced and vestigial eyes lacking superposition or apposition structure.

Johnson et al. (2015), *PeerJ*, DOI 10.7717/peerj.1176

**Table 1  Eye types and nebenaugen by family.** Classification follows *De Grave & Fransen (2011)*, although Acanthephyridae and Oplophoridae are considered the same family, following *Wong et al. (2015)*. The distribution of nebeaugen and eye types amongst caridean families. Eyes are split into 6 types as described in Table 1 and Fig. 1. The binomial test results indicate whether the distribution of nebenaugen deviates significantly from 50:50 by group or eye type.

| Family | 1–straight | | 2–sinuous | | 3–forward | | 4–large | | 5, w-shape | | 6, other | | All | | Binomial test |
|---|---|---|---|---|---|---|---|---|---|---|---|---|---|---|---|
| | NE | No NE | NE | No NE | NE | No NE | NE | No NE | NE | No NE | NE | No NE | NE | No NE | |
| Alpheidae | | 2 | | | | | | | | | | 34 | | 36 | $P < 0.0001$ |
| Anchistioididae | 1 | | | | | | | | | | | | 1 | | |
| Atyidae | | 7 | 1 | 6 | | 1 | | 1 | | | | | 1 | 15 | $P < 0.001$ |
| Barbouriidae | | | | 1 | | | | | | | | | | 1 | |
| Campylonotidae | | | | | | | | | | 1 | | | | 1 | |
| Crangonidae | | 1 | | 3 | | 1 | | 2 | | 8 | | 3 | | 18 | $P < 0.0001$ |
| Desmocarididae | | | | | | | | | | 1 | | | | 1 | |
| Disciadidae | | 1 | | 1 | | 1 | | 1 | | | | | | 4 | |
| Eugonatonotidae | | | | | 1 | | | | | | | | 1 | | |
| Euryrhynchidae | | | | | | | | | | | | 1 | | 1 | |
| Glyphocrangonidae | | | | | | | | 1 | | | | | | 1 | |
| Gnathophyllidae | 1 | 1 | | | | 1 | | | | 1 | | 1 | 1 | 4 | |
| Hippolytidae | 2 | 9 | 10 | 6 | | 3 | | 1 | | 1 | | 2 | 12 | 22 | n.s. |
| Hymenoceridae | 1 | | 1 | | | | | | | | | | 2 | | |
| Nematocarcinidae | | | | 3 | | 1 | | | | | | | | 4 | |
| Ogyrididae | | 1 | | | | | | | | | | | | 1 | |
| Oplophoridae | 1 | | 3 | 1 | 2 | 2 | | | | | | 1 | 6 | 3 | n.s. |
| Palaemoninae* | | | 14 | 1 | | 1 | | | | 1 | | 1 | 14 | 4 | $P < 0.05$ |
| Pandalidae | 1 | 6 | 6 | 4 | | | 5 | 2 | | | | | 12 | 12 | n.s. |
| Pasiphaeidae | | 3 | | | | 1 | | | | 2 | | 1 | | 7 | $P < 0.05$ |
| Pontoniinae* | 23 | 59 | 6 | 6 | 4 | 5 | | 2 | | 5 | | 1 | 33 | 78 | $P < 0.0001$ |
| Processidae | | | | | | | 1 | 4 | | | | | 1 | 4 | |
| Rhynchocinetidae | | | 2 | | | | | | | | | | 2 | | |
| Stylodactylidae | | | | 1 | | 1 | | | | | | | | 2 | |
| Thalassocarididae | | | | | | | 1 | | | | | | 1 | | |
| Xiphocarididae | | | | 1 | | | | | | | | | 1 | | |
| Total | 30 | 90 | 44 | 33 | 6 | 18 | 7 | 10 | 1 | 24 | 0 | 50 | 84 | 224 | $P < 0.0001$ |
| Binomial test | $P < 0.0001$ | | n.s. | | $P < 0.05$ | | n.s. | | $P < 0.0001$ | | $P < 0.0001$ | | | | |

**Notes.**

* Palaemonidae

NE, nebenauge.

because it was favoured by the most speciose sub-family, the Pontoniinae. Fifteen families possessed the next most common, Type 2 eyes which is the only classification that has more species with nebenaugen than without. Type 6, which generally consisted of reduced eyes, was the only classification where none of the examples examined possessed a nebenauge.

## Nebenaugen

We examined the eyes of 308 non-troglobitic genera from 26 families (Table 1) and found that 88 possessed a discernible nebenauge (28.6%). We found that the distribution of nebenaugen across caridean families, where we sampled more than the minimum of 5 genera, deviates from a 50:50 distribution overall ($p < 0.001$, Table 1). There appear to be some links between its presence/absence and systematics with non-random differences in the prevalence of nebenaugen. The distribution of nebenaugen amongst genera by family could be split into those where genera had about a 25–50% chance of having one (Hippolytidae, Oplophoridae and Pandalidae), those where they occurred infrequently (Pontoniinae and Atyidae) and those where none were found for any species (Alpheidae and Crangonidae). The only family where significantly more genera were found to have nebenaugen than not were the Palaemoninae. There are no families where all genera possess a nebenauge. Analysis of the data for the 7 families where there were more than 10 genera sampled suggested that the pattern was not random ($X^2 = 53.68$, d.f. $= 6,245$, $p < 0.001$). For families where fewer than 10 animals were sampled, the proportion of those in possession does not deviate significantly from 50:50 (i.e., $p > 0.05$) with the exception of the Pasiphaeidae ($n = 7$). Given the small numbers in these families, however, it would perhaps be unwise to draw any conclusions.

A chi-squared test of distribution of nebenaugen between commensal and free-living genera indicated that there was no deviation from independence ($X^2 = 1.0712$, d.f. $= 1,308$, $p = 0.399$). No deviation from independence was noted for the distribution between freshwater and marine genera ($X^2 = 0.0136$, d.f. $= 1,303$, $p = 0.9073$) or by depth class ($X^2 = 1.012$, d.f. $= 3$, $p = 0.799$). Neither was there any a significant pattern noted in relation to the light regime shrimps experienced; a similar prevalence of nebenaugen was noted for deep, temperate and tropical-living decapods ($X^2 = 1.297$, d.f. $= 2$, $p = 0.523$). However, the distribution of nebenaugen among shrimps depending on the size class of their rostrum (Fig. 3) appears to show a pattern ($X^2 = 32.583$, d.f. $= 1,256$, $p < 0.001$). There is a positive association between the likelihood of a shrimp with a particular eye type having a medium or long rostrum and it having a nebenauge (Kendall rank correlation, tau $= 0.867$, d.f. $= 1,5$, $p = 0.01667$).

## Allometry of the eye and nebenauge

Neither carapace length nor eye diameter were normally distributed and so were logged. There is a significant relationship between log carapace length (CL) and log eye diameter ($p < 0.0001$, $r^2 = 0.656$, d.f. $= 1,168$). A least squares regression gave the relationship log eye diameter $= 0.723$ log CL $- 1.315$ (Fig. 4). Overall, animals with nebenaugen tended to have larger eyes both in terms of absolute dimension ($F = 6.449$, d.f $= 168$, $p = 0.012$) and relative to carapace length ($F = 5.997$, d.f. $= 168$ $p = 0.0154$) than those that did not.

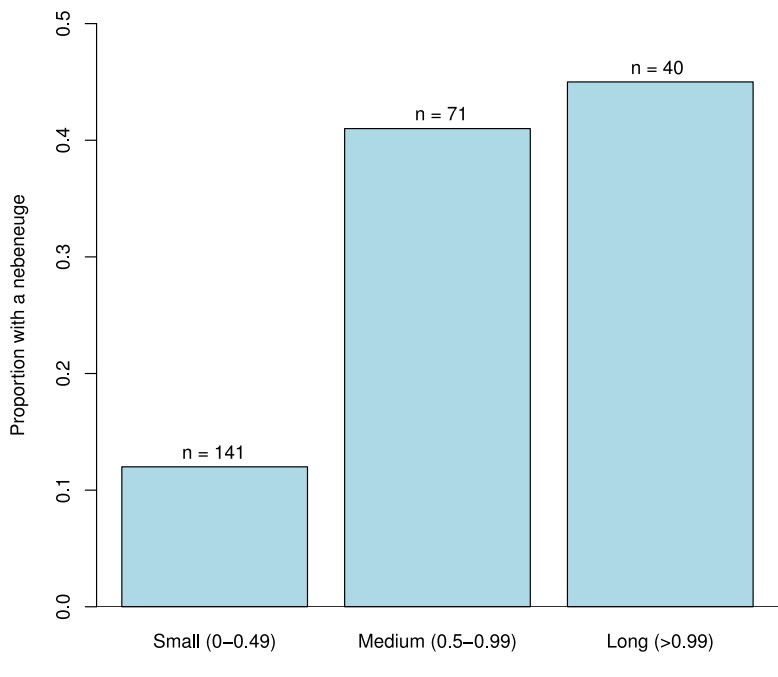

**Figure 3 Nebenaugen proportions.** Proportion of caridean decapods that have a nebenauge in relation to the relative length of their rostra. Rostrum size is relative to the carapace length.

Although no significant difference in relative eye diameter could be demonstrated within families, the tendency for larger animals to have nebenaugen was consistent among the five families where there were sufficient species with and without nebenaugen (Fig. 5). Also, a comparison of relative eye diameter among rostrum size classes (Fig. 6) revealed that animals with larger rostra relative to carapace length tended to have larger eyes (Kruskal-Wallis, $X^2 = 19.193$, $p < 0.0001$).

When the size of the eye relative to carapace length was examined according to broad habitat and depth classifications (Fig. 7) there was no variation for species that had a nebenauge (KW test, $X^2 = 0.589$, $p = 0.899$) but for those lacking nebenaugen there was significant variation (KW test, $X^2 = 13.817$, $p < 0.005$). This was caused by the significant difference in relative eye diameter between deep living species depending on the presence/absence of the nebenauge (Wilcox test; $W = 58$, $p < 0.005$). Examination of the angle over which the nebenaugen occupied the dorsal surface of the eye suggested that there was a significant difference by marine habitat classification (Kruskal-Wallis, $X^2 = 9.5025$, df $= 3$, $p$-value $= 0.0233$) with freshwater animals generally having much reduced nebenaugen (Fig. 8). The nebenaugen angle in species that are known to undergo diurnal vertical migration (i.e., oplophorids, pandalids, pasiphaeids; *Aguzzi & Company (2010)*) is significantly larger than in those that do not (Wilcox $W = 65$, $p < 0.001$).

## DISCUSSION

We have demonstrated that in caridean decapods eye size has an approximately log–log relationship with carapace length (Fig. 4). We have also found that there is a general trend for

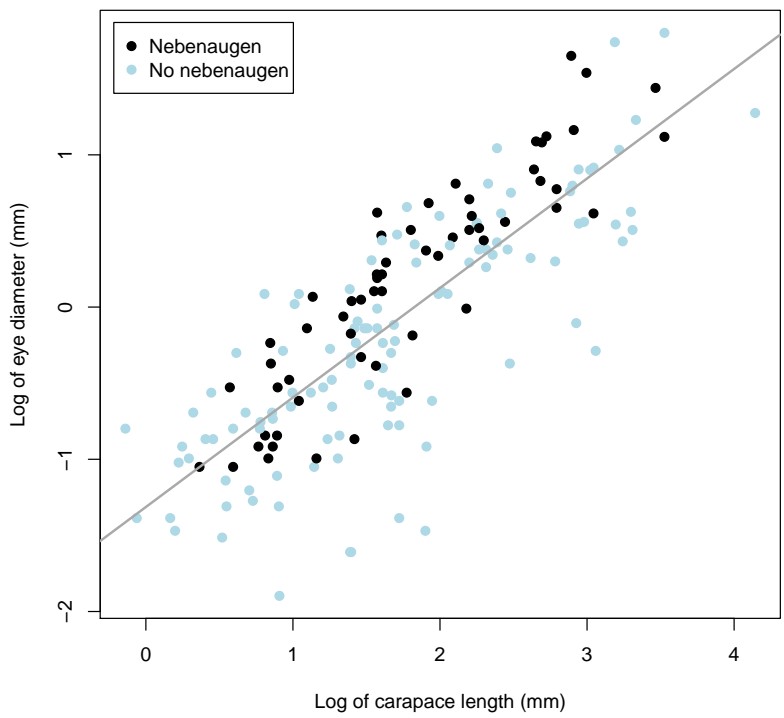

**Figure 4 Carapace length v eye diameter.** The relationship between log carapace length and log eye diameter in caridean decapods.

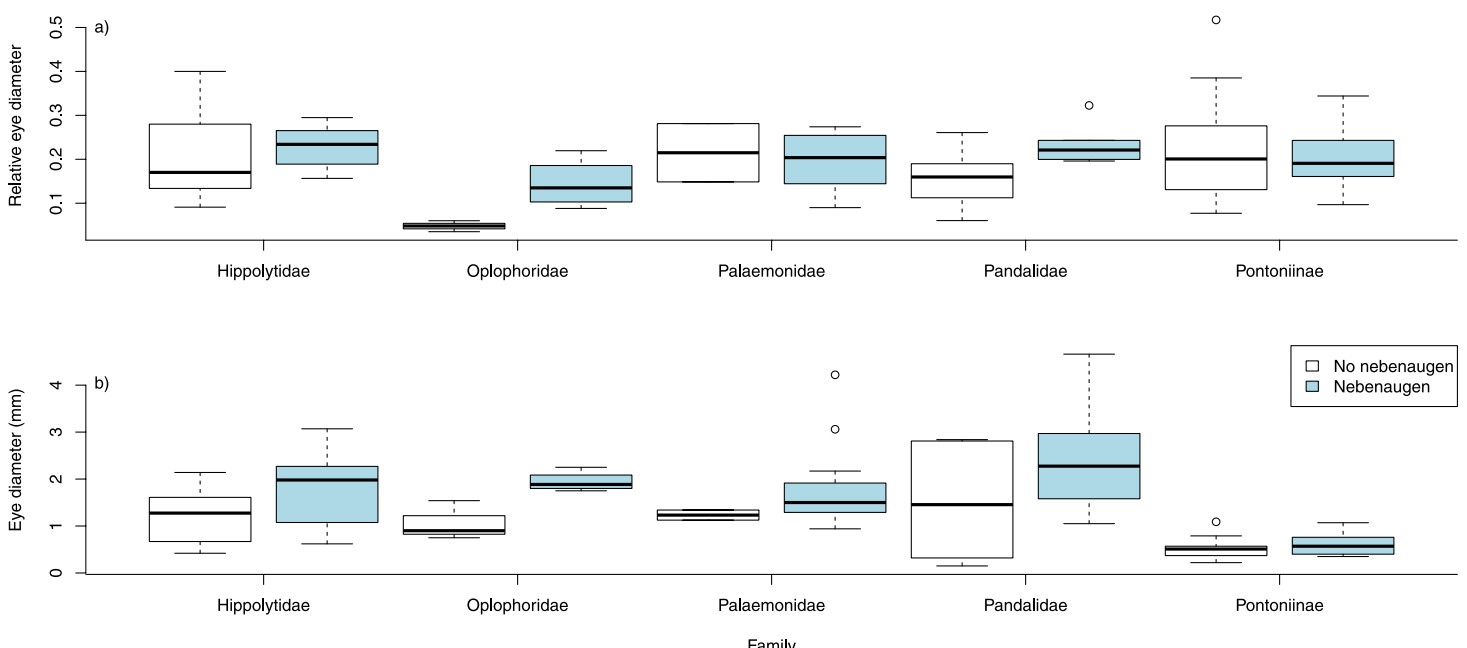

**Figure 5 Relative eye diameter and nebenaugen.** Relative eye diameter in relation to the presence or absence of a nebenauge. Animals with nebenaugen generally have larger eyes relative to carapace length.

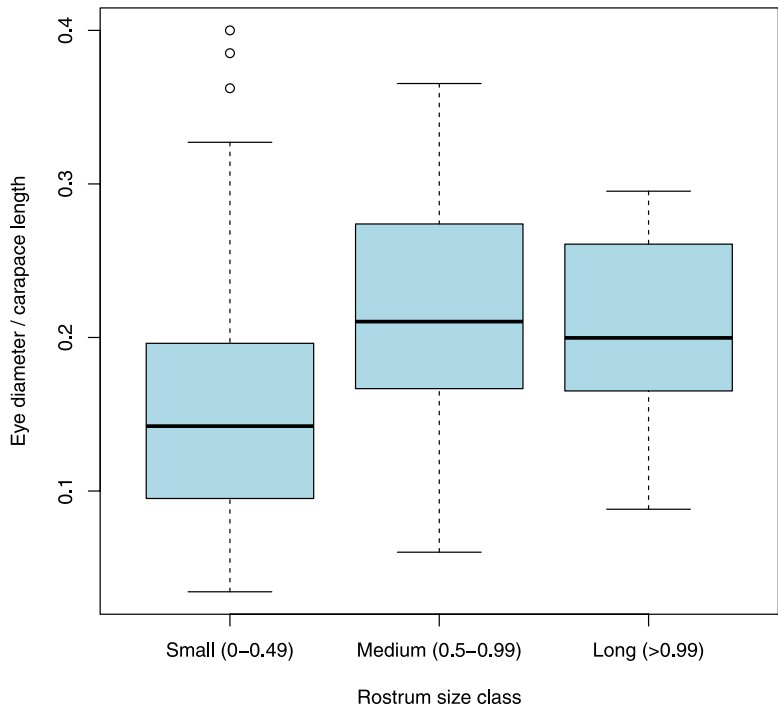

**Figure 6 Eye diameter and rostrum length.** Comparison of relative eye diameter between animals that have a rostrum less than or greater than the length of their carapace.

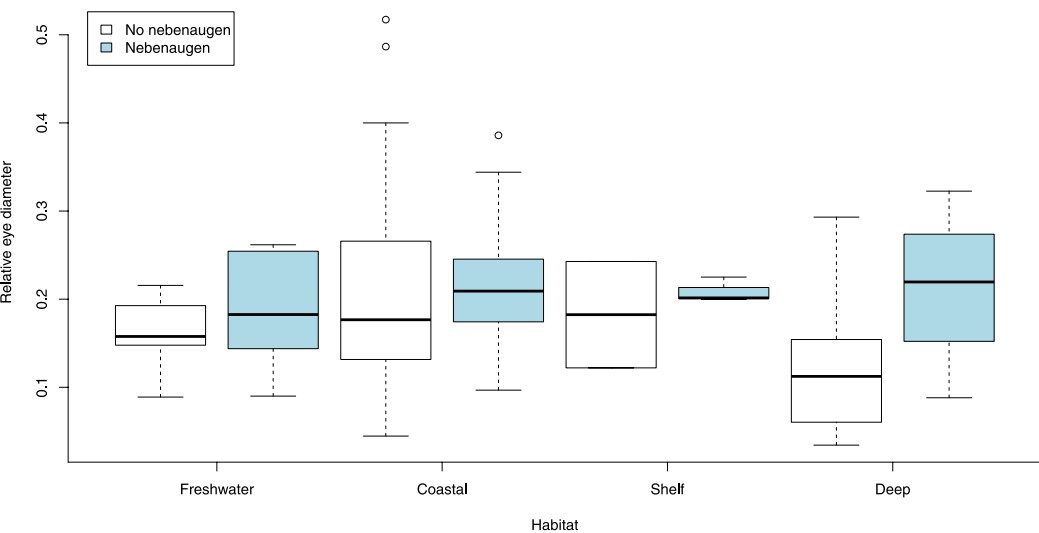

**Figure 7 Eye diameter and depth.** Relative eye diameters of animals from freshwater and three marine depth ranges depending on presence/absence of nebeaugen. Depths are Coastal 0–100 m, Shelf 101–300 m, Deep, >300 m.

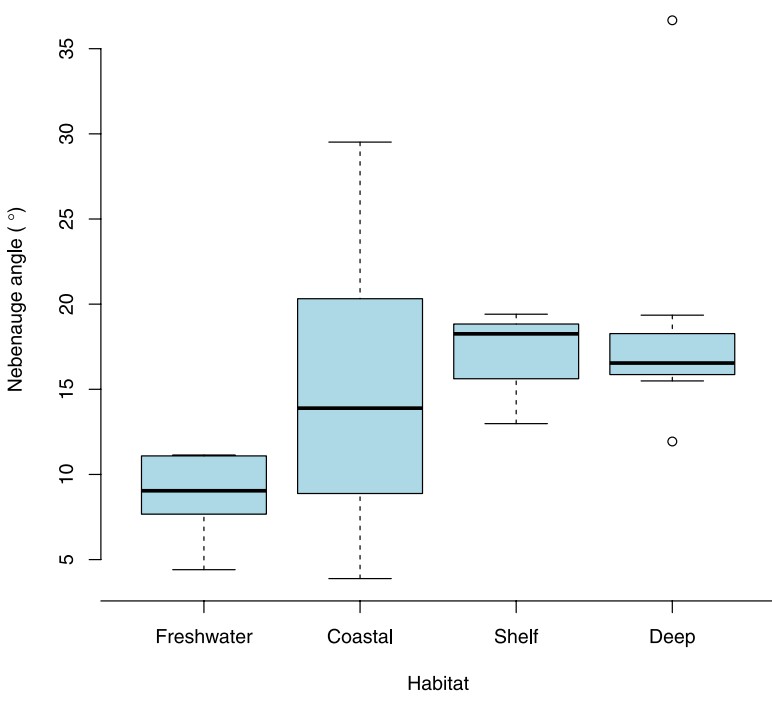

**Figure 8 Nebenaugen angle and habitat.** Comparison of the angle that nebenaugen occupy by habitat (angle calculated as 360 (nebenauge diameter/eye circumference)). Depth ranges are Coastal, 0–100 m; Shelf, 101–300 m; Deep, <300 m.

animals that have nebenaugen to have larger eyes relative to their carapace length. Larger eyes can either deliver greater sensitivity through a larger aperture and larger rhabdoms or offer an opportunity for better resolution through reduced interommatidial angles.

Although the link may be considered tenuous given the numbers of pelagic decapods that have small rostra (*Sarda, Company & Costa, 2005*), we have used the presence of a large rostrum, which would be a physical hindrance to an endocommensal or infaunal habit, as an indicator that these species are likely to be free-living and active swimmers. Our data suggest that animals that have a long rostrum relative to carapace length tend to have larger eyes and are more likely to have nebenaugen (Figs. 3 and 6). These findings concur with a previous study focussed on symbiotic pontoniine decapods (*Dobson, De Grave & Johnson, 2014*) where it was found that endosymbiotic species had smaller eyes and were less likely to have nebenaugen. The costs and benefits of having large eyes will be a function of various ecological parameters such as temporal activity patterns, mode of prey capture and the availability of visually relevant light (*Talarico et al., 2007*; *Rutowski, Gislén & Warrant, 2009*). Offset against any potential advantages, the rapid rise of metabolic cost with increasing size in compound eyes is likely to encourage the evolution of the smallest eye appropriate to the needs of the animal (*Weibel, 2000*; *Laughlin, 2001*).

In marine species, the likely relationships between habitat, lifestyle, gross eye morphology and eye size will be further complicated by consideration of habitat depth. Depth has been shown to have a strong impact on the morphology and function of decapod eyes

(*Johnson et al., 2000*). Animals found in shallower water tend to have "faster" eyes with a greater capacity for light adaptation than those from the deep sea (*Johnson, Shelton & Gaten, 2000*). With regard to eye diameter, our finding that for species without nebeaugen relative eye size is smallest in deepsea species is contrary to that of *Hiller-Adams & Case (1985)* for benthic decapods but matches that of *Hiller-Adams & Case (1984)* for euphausiids. This may reflect the fact that many smaller decapods have a demersal rather than truly reptant or benthic lifestyle, more similar to euphausiids than lobsters. The general pattern in these decapods appears to be that, at greater depths, eyes become less important; this is perhaps a reflection of some suggestions that the relative importance of chemosensitivity is greater in deep-sea species (*Fuzessery & Childress, 1975*; *Marshall, 1979*). The decrease in eye size may correlate with observations by neuroanatomists of changes in brain architecture in some caridean decapods that may indicate a reduction in visual capability in species with small eyes (*Sandeman, Scholtz & Sandeman, 1993*). Bathypelagic fish have small eyes with large pupils, which *Warrant (2000)* suggests is a reflection of the changing nature of the visual scene with depth—animals at depths beyond daylight penetration are adapted to see point sources of light rather than broad and detailed visual scenes.

For species that have a nebenauge, our data suggest that there is no trend in eye size with depth-related habitat classification (Fig. 7) but that for these species the organ is as large or larger in species generally found below 100 m (Fig. 8). This new finding suggests that the organ is an important adaptation for deep sea species, many of which in our study are regarded as pelagic. Although food availability is implicated in vertical migrations of some species such as *Euphausia superba* (*Tarling & Johnson, 2006*), it is generally accepted that light triggers diurnal vertical migration of planktonic organisms (either directly or indirectly) in temperate and tropical regions (*Aguzzi & Company, 2010*; *Ochoa et al., 2013*). The large size of the nebenaugen when it is present in species found below 100 m or that are mesopelagic vertical migrators may lend further credence to the suggestion by *Gaten, Shelton & Herring (1992)* that the organ functions as a photometer for monitoring downwelling light and is linked to the control of diurnal vertical migration.

We have found that the prevalence of nebenaugen within families is not random. Of the families that lack nebenaugen, the Alpheids are burrow/shelter dwellers, the Atyidae are mostly freshwater, the Crangonidae are highly benthic and burrowing (*Bauer, 2004*) and the Pasipheidae have smaller eyes relative to their carapace length than all families except the Ogyrididae. Of the families where nebenaugen are common, the Oplophoridae are meso and bathy-pelagic, Hippolytidae are diverse epibenthic marine perchers, Palaemoninae are a diverse sub-family that has repeatedly invaded freshwater, Pandalidae are generally medium to large epibenthic schooling species and Pontoniinae are exclusively marine/brackish water, hugely diverse and often symbiotic (*Bergstrom, 2000*; *Bauer, 2004*; *Ashelby et al., 2012*). These factors lend credence to the idea that the presence and absence of the nebenauge in carideans is loosely tied to lifestyle; active epibenthic or pelagic species are more likely to have them than burrowing mostly benthic species. However the systematic or indeed ecological pattern of occurrence is not absolute.

We have concluded that there are six recognisable types for caridean decapod eyes based on the shape of the eye and the boundary between the cornea and eyestalk, and noted that the distribution of nebenaugen is also not random among eye types. Animals with type 6 eyes, such as the Alpheidae, never have a nebenauge. For 4 out of the 6 eye types examined, the prevalence of the nebenauge is less frequent than would be expected by chance, and for the other two families the distribution does not differ significantly from 50:50. Given the current uncertainty as to how the nebenauge is connected to the x-organ/sinus gland (*Carlisle, 1959*) there is a possibility that the distinctions we have made between eye types may be cosmetic as much as functional. If the nebenauge has an important function, the obvious questions are: (1) What does it do; and (2) How is its function covered in species that do not have one? Given the inconsistencies in systematic and ecological distributions of the nebenauge, it seems likely that there is no single simple answer—it may have different functions in different groups. However, given the plasticity of systems determining endogenous rhythms (*Yerushalmi & Green, 2009*) and the possible links the nebenauge has with the sinus gland complex, this area merits further investigation.

## CONCLUSION

We conclude that, although as with previous investigations there appears to be no clear systematic pattern in its distribution by family, and although we cannot ascribe a function to the nebenauge of caridean decapods, its presence appears to be linked to lifestyle. It is more frequently present as the eye size of a species increases, and shows an association with relative rostrum length. If size is an indication of ecological importance, then the fact that it is larger in species found below 100 m suggests that it has a particularly important role in some deep-water species. With regard to the superficial morphology of caridean eyes, we have found a log–log relationship between carapace length and eye diameter, and suggest that there are 6 identifiable eye morphotypes based on general morphology and the margin between eyestalk and cornea.

## ACKNOWLEDGEMENTS

We gratefully acknowledge the detailed comments of B Schoenemann and VB Meyer-Rochow, which markedly improved the manuscript.

### Funding

The authors gratefully acknowledge the St Johns College, Oxford, senior summer scholarship granted to MLJ, which facilitated his access to the Oxford University Natural History Museum, and funding from the Systematics Research Fund to SDG (jointly administered by the Linnean Society and the Systematics Association). The funders had no role in study design, data collection and analysis, decision to publish, or preparation of the manuscript.

## Grant Disclosures

The following grant information was disclosed by the authors:
St Johns College, Oxford.
Systematics Research Fund.

## Competing Interests

Magnus Johnson is an Academic Editor of PeerJ.

## Author Contributions

- Magnus L. Johnson conceived and designed the experiments, performed the experiments, analyzed the data, contributed reagents/materials/analysis tools, wrote the paper, prepared figures and/or tables, reviewed drafts of the paper.
- Nicola Dobson conceived and designed the experiments, performed the experiments, contributed reagents/materials/analysis tools, reviewed drafts of the paper.
- Sammy De Grave conceived and designed the experiments, contributed reagents/materials/analysis tools, wrote the paper, prepared figures and/or tables, reviewed drafts of the paper.

## Supplemental Information

Supplemental information for this article can be found online at http://dx.doi.org/10.7717/peerj.1176#supplemental-information.

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
