# Peer review of "External morphology of eyes and Nebenaugen of caridean decapods–ecological and systematic considerations"

_PeerJ, doi:10.7717/peerj.1176_

## Round 0.1 · original submission · Major Revisions

Dear Authors,

Thank you for submitting your very interesting article. All the reviewers found merit in the data presented but have suggested a number of revisions prior to publication.

Reviewer 1 ·

Basic reporting

This paper represents the most complete examination to date of a little studied field – the eyes and nebeneuge in crustaceans. It is well written and the findings are appropriately related to previous work on the subject.

Experimental design

A considerable number of specimens have been examined in appropriate detail and clearly described.

Validity of the findings

The number of specimens used us more than adequate and the findings have been analysed using statistics that appear to be sound and well used.
The results presented here do not offer a definitive answer to many of the questions, but that relates more to the intractability of the problem than to any deficiencies in the research.

Additional comments

Line 200 states ‘Larger eyes generally indicate increased sensitivity and resolution through a larger aperture, longer rhabdoms and decreased interommatidial angle’. Larger eyes occur for two different reasons. They may be larger to increase sensitivity via a larger aperture and longer rhabdoms, or they may be larger to increase resolution via decreased interommatidial angle and more rhabdoms. These two properties are very different and often mutually exclusive, and they should not be combined in a single sentence like this.

Line 217. I do not like the term ‘match’. The absence of the tapetum in parts of these eyes is related to the likelihood that reflected eyeshine would reveal the animal to potential predators. The presence or absence of tapetal pigment does not match the continuously varying downwelling light, although it is clearly related to it.

A few minor points:

Line 48: delete ‘no’. Should be ‘is of relevance to vision’.
Line 49: tapetal distributions do not match the angular distribution of light.
Line 348: ‘irrandiance’ has a surplus n in it.
Fig 1a: could we have an arrow marking the DS. It is not clear which where it is precisely.
Fig 3 legend refers to 'dorsal spot' and to ‘nebeaugen’.
Fig 5c legend. The degrees that the dorsal spot occupies…. Do they mean the angle?

·

Basic reporting

Figure 1a is not clear

Experimental design

is ok

Validity of the findings

-

Additional comments

Review (PD Dr. B Schoenemann, University of Cologne):
Nebenauge of carideans external morphology of eyes and the nebenauge of caridean decapods – ecological and systematic considerations
Magnus L. Johnson, Nicola Dobson, Sammy De Grave

This seemed to be a very promising manuscript, and all the more as hardly anything is known about these puzzling Nebenaugen of decapod crustaceans – it sells itself, however, a bit under value.

Nebenaugen
The main reason for my concern is that quite a number of parts of the texts , as for example lines 70-72, 79-82 are not well-phrased. The terminology used is confusing. Obviously the Nebenauge is not an ocellus but a kind of compound eye. (Is anything known about their internal morphology?) An ocellus is morphologically very different from a compound eye. The ocellus consists just of a small lens with a small retina below. In crustaceans the dioptric apparatus is reduced or can be missing. Ocelli can be found as median eyes in insects (as a second visual system besides the compound eyes), as Naupliusauge in crustaceans, but also in some adult crustaceans (comp. Koenemund & Jenner 2005) or as the main system as larval eyes of insects such as caterpillars, here called stemmata. Typically in adult crustaceans the ocelli disappear. The ocelli which can be found in myriapods, forming aggregate eyes, or in spiders, are formed secondarily from ommatidial compound eyes. So in this context the term ocellus, pigmented ocellus, dorsal eye (sometimes used also for median eyes), should not be used. The same is valid for the term spot eye, eye spot, pigmented spot or similar expressions. (An eye spot is a tiny assemblage of light sensitive cells isolated from each other by pigment cells. These small organs are just light detectors.)
At several parts of the text the Nebenauge is directly compared to the ocelli of insects (e.g. 89-90). I wonder whether this makes sense in view of what I have said before. If there is seen any homology between ocelli and Nebenaugen, this must be explained in the text, and would be very important in this and in an evolutionary context. If there is no homology, these comparisons should be taken off. Other lines where this is relevant: 232 (eye spot), 80, 246, 247-48
I very much would like to get to know more about these Nebenaugen, which, when translated from German to English would effectively mean “associated eyes”. In Figure 1a the relevant area hardly can be seen, Figure 1b is ok, but one really would be interested to see more types of these Nebenaugen addressed in the text. This is all the more important since figure 2 is, I am sorry to say that, a bit chaotic. It shows the different types of stalked eyes, which is nice, but hardly helps with the Nebenaugen, and the figure appears in its construction not well ordered. This should be improved.

Aims
Troglodytic genera. It might have been interesting to know whether these forms had any eyes, which types …


Analysis (Methods)
Please clarify what R is (line 136).

Nebenauge
I know what you mean, but wonder as formulated, whether the 50 : 50 question is an appropriate form to describe which genus has Nebenaugen and which has not. In the following questions it makes more sense, of course.
Logic: It is stated that the probability of finding Nebenaugen increases with the size of the compound eyes, and that in caridean crustaceans the eyes become smaller the deeper the region in which they live. In line 169 it is said that depth is not relevant. I see here a contradiction. Please clarify.
Possibly one even could speculate that if the Nebenaugen can be found just in regions closer to the surface or lower depths, they need more light to function.

Discussion
One would like to know, why crustaceans with a short rostrum are thought to be more active, or pelagic, even if it refers to another author (Sarda 2005). Please add some information.
246-248 really dorsal ocelli? (see above)
Quite a number of points are discussed here – and it should be more acutely should be pointed out, what has been found. In my view there are three points:
- The occurrence of Nebenaugen is independent of many factors such as ….
It is important not to be negative here but to present it as a positive result.
- A general trend towards the existence of Nebenaugen increases with increasing size of the eyes and rostrum while the size of the eyes decreases with depth.
- The presence of Nebenaugen seems thus linked to an active life-style
- The link to the neurosecretory glands and the idea of some relevance for determination of endogenous rhythms seems important…

- Facit: Probably Nebenaugen are of use for demersal caridean decapods.


Please reconstruct the discussion, and make your results sharper. Principally: after sentences like : Our data suggest … please mention which data, for example fig. xy. (e.g. line 204)


Further Amendments:

Title: Nebenauge (not nebeneuge)
19 Doppelauge (not doppleugen)
23 I don´t see the relation of Darwin´s sentence and the context. (What have the physical constraints to do with “immutable contrivances”?
61 1938) who please check the whole manuscript for double spaces
66 The Nebenauge, not: The Nebeneuge
80 … structure of the ocellus resembles that of the eye proper … this is probably wrong, because it is not an ocellus
89 … Be careful to compare the insect ocellus with this system discussed here
107 presence/absence
122 Why “although”?
140-150 a bit hard to follow….
174 particular – which? please clarify
192 (Kruskal … not ((Kruskal …
202 active or pelagic is not a good construction, comparison. It is possible to be active and pelagic.

Text and figures for dorsal spot in decapods: avoid "spot"

Figures and tables
4 caridean not Caridean (if consistent with the rest of the text)
7 aren´t there 6 types? (Mention the reduced forms)

Fig 3 twice: nebeneuge instead of Nebenauge in the scales of the diagrams and in the box
9 avoid “dorsal spot”
11 nebeneuge instead of Nebenauge

Fig 4 Explain “relative” eye diameter – relative to what?
Explain the description of your figure. … in relation to the presence (Y) or absence (N) …
13 nebeneugen instead of Nebenaugen

Sg. Nebenauge, Pl. Nebenaugen

·

Basic reporting

Please read Referee's Report:

Some major revision is required regarding background information on the problem of photoreceptor adaptations in crustaceans generally and the 'nebenauge' in particular.
The Introduction contains material that had better be in the Discussion and references to single-lens eyes in verterates and spiders confuse the issue and take away attention from the problem of the nebenauge and compound eyes. The idea of comparing adaptations across phylogenetically related but ecologically diverse species is a very good and useful one.

Experimental design

All of the observations were carefully carried out on ethanol infiltrated specimens, but the duration the specimens have spent in the alcohol (as well as possible sexual dimorphic differences related to eye designs) are not known. This could have affected the measurements and some the conclusions.

Validity of the findings

The validity of the findings is fine, but the findings do not warrant a discussion as extensive as the one in this manuscript. The paper could have been shorter and focusing on crustacean eyes (rather than have fish, bird and spider eyes entered into the discussion, which is unwarranted and in fact counter-preoductive).The authors try to make a little "too much" out of their study and in doing so overhoot their aim. Unfortunately they miss many key-papers on crustacean eyes and especially studies on compound eye adaptations to different environments. They also need to define clearly what a nebenauge is vis-a-vis frontal eyes, dorsal ocelli, intracerebral ocelli, stemmata etc.

Additional comments

Referee’s Report on the manuscript “External morphology of eyes and nebenauge of caridean decapods – ecological and systematic considerations” by GM. L. Johnson, N.C. Dobson. Buscaino et al. and S, DeGrave

Overviews and comparisons between ecologically diverse, but phylogenetically closely related species are always extremely useful for physiologists in particular and zoologists generally and I’d be happy to see this manuscript published once the authors have revised it thoroughly and heeded some of my suggestions. Some very useful correlations between rostrum length, for example, and nebenauge-presence in combination with larger eyes (the authors probably mean the main compound eyes or do they also refer to the size of the nebenauge?) have emerged.

There are, however, several areas that need to be addressed and in places the text is somewhat verbose, even repetitive, and could be improved if condensed. Furthermore the various comparisons or references to single lens eyes, be they those of vertebrates (like fish and birds) or those of insects and spiders (like ocelli), are distracting and not really justified. Instead the paper should focus on compound eyes for which a vastly more numerous and relevant literature (not cited by the authors) exists than what the authors have referred to (more on this further below). In the following I shall go through the manuscript from start to end, commenting on sections that I feel could be improved as they come along.

Abstract, line10: First sentence. Better write “Most caridean decapods have compound eyes of the reflecting superposition kind and some additionally possess an accessory ocellus or nebenauge of unknown function”.
If “most” caridean decapods have reflecting superposition compound eyes, the reader will wonder what kind of photoreceptors those who do not have this type of eye possess and why they do not have the most common type of eye present in the taxon. Furthermore it is probably wrong to equate the nebenauge with an ocellus (the latter in crustaceans given a variety of names and even thought to include intracerebral ocelli, cf. Gilbert Martin and co-workers in France; Bobkova et al.).
What is “an active life style”? Chasing food or swimming around more actively for other reasons? Would female and male individuals always lead the same “lifestyles”? There are interesting studies on the eyes of sexually dimorphic insects in which males and females occupy different ecological niches, e.g. flying males with large superposition eyes and wingless, largely sessile females with smaller eyes (e.g., some fireflies and moths).

Introduction, line 21 etc.: Not only the laws of physics “impose regulations on the design of eyes”, but the biochemistry of vision cannot be ignored. Physics is involved with regard to optics and size constraints.
L23: Has Darwin ever considered compound eyes and in the light of today are eyes generally really “immutable contrivances”? As far as I know his conclusions were based on single-lens eyes, but perhaps his statement refers to all kinds of photoreceptors.
L33: An important and more recent review than that of Cronin 1986 on crustacean compound eyes by Meyer-Rochow (2001 in Zool. Sci.) has been missed and should be included.
L35: “…thought to be an early superior development of apposition optics…” That is nowadays questioned and even intermediate eye types between superposition and apposition have been described.
L37: Write “…quite different from an apposition eye..” (instead of “to”)
L39,40: The authors need to consult the review on conpound eye adaptations in arthropods that live in extreme conditions (Meyer-Rochow & Nilsson 1999 in Springer Verlag book edited by Eguchi & Tominaga). Bioluminescence is most common in the mesopelagial; at even greater depths it becomes increasingly rare.
L. 41-54: Much of this had better be part of the Discussion.
L. 55-65: The references to bird vision are unjustified, for single lens eyes operate on totally different principles from those of superposition compound eyes. References to Welsh & Chace (1937, 38) should be replaced with the more up-to-date review by Meyer-Rochow and Nilsson (1999) and/or the study by Eguchi et al (1997, Biol Bulletin).
L. 67-78: In the brief description of the nebenauge it should be made absolutely clear how it can be morphologically distinguished from ocelli, frontal organs, intracerebral ocelli etc. in terms of size, position and appearance (cf. Meyer-Rochow 1999 in Springer Verlag book edited by Eguchi and Tominaga).
L. 79-99. Much of that is Discussion. If eyestalk removal were to be discussed at all (is that relevant to this study of museum specimens?) David Sandeman’s results on eyestalk ablation and subsequent appearance of an additional antennule in place of the eye should be mentioned.
L. 100: OK.

Methods, L.111: Could the 75% ethanol and the duration the specimens spent in it have had an effect on size and shape of the eyes, given the phase within the moulting cycle the specimens were in when they were fixed?
L.122: By which criteria do the authors judge that an eye is a “functional superposition eye”? How to distinguish a “functional’ from a “non-functional” superposition eye?
L.122: A “troglodyte” is a human cave-dweller ! The authors mean either “troglobite” (i.e., an animal that occurs exclusively in a cave) or “troglophile” (an animal that prefers to live in a cave, but can also survive outside the cave).

Results, L. L.152: Wrong word “non-troglodytic” must be replaced. Troglodytic refers to people living in caves (not animals) !
Assuming the statistics are correct (and the massive amount of data, i.e., measurements of course did of course require such treatments), one still wonders if the different lengths of time the specimens have spent in the alcohol could not have affected the results.
L. 153: Here and elsewhere, use the definitive article with nebenauge “...distribution of the nebenauge…”; L 156 “…of the nebenauge…”

Discussion: Once again references to vertebrate eyes (being single lens eyes) or spider eyes are not helpful, but a discussion on superposition size constraints, e.g., that Meyer-Rochow & Gal (2004) in Vision Research is essential in this context.

L.212-230: The discussion between lines 212 and 230 on tapeta in the nebenauge is far too extensive and repetitive and should be combined with information already given in the introduction and, once again, reference to bathypelagic fish (which incidentally may even lack eyes altogether) is unwarranted. The results say little or nothing on the presence or absence of tapeta in the nebenauge.
L. 231-244: This referee is not at all happy about this part of the Discussion. The nebenauge is confused with eyespots and even dorsal ocelli in insects (which are totally different in terms of ontogenetic origin, structure and presumably function). Referring to them removes the focus on the nebenauge and confuses the issue further. The authors should remember that they have no evidence to conclude anything on the function of the nebenauge based on their morphological measurements of alcohol-preserved material. Parry (1947) is an out-of-date reference with an incorrect statement: there are many insects that are poor flyers or even lack wings, but possess ocelli.
L. 243: Write “….that the nebenauge is more prevalent…”

References: Some very relevant and important ones related to the topics of ecophysiological adaptations and photoreceptor plasticity have been missed; less relevant ones and some outdated ones ought to be dropped.

Figures: Acceptable on the whole and informative.

---

## Round 0.2 · accepted · Accept

Thank you for resubmitting your revised version of your manuscript. I'm delighted to accept this paper and thank you once again for submitting your research to PeerJ.